# GRIT: Teaching MLLMs to Think with Images

**Yue Fan**[1]    **Xuehai He**[1]    **Diji Yang**[1]    **Kaizhi Zheng**[1]
**Ching-Chen Kuo**[3]    **Yuting Zheng**[3]    **Sravana Jyothi Narayanaraju**[3]
**Xinze Guan**[3]    **Xin Eric Wang**[1,2]
[1]UC Santa Cruz    [2]UC Santa Barbara    [3]eBay
yfan71@ucsc.edu, ericxwang@ucsb.edu
https://grounded-reasoning.github.io

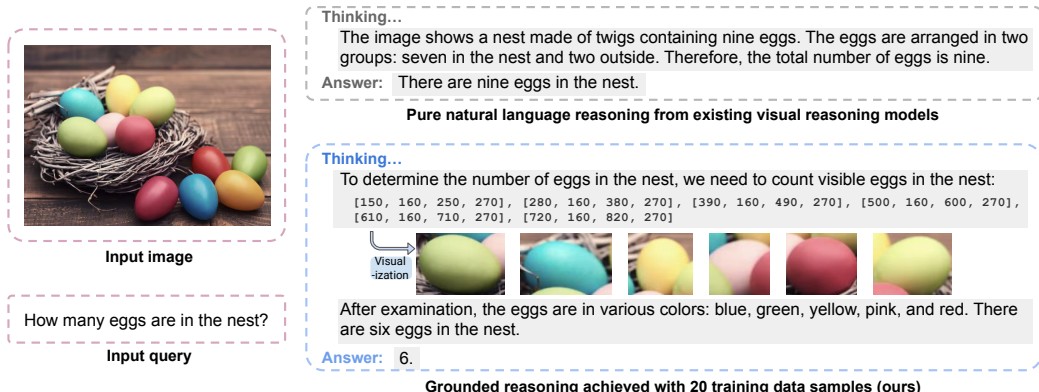

Figure 1: Comparison of reasoning with pure natural language and our grounded reasoning that mixes explicit bounding boxes for image regions with a chain of natural language thoughts. Our GRIT method enables MLLMs to perform grounded reasoning with only 20 training samples, realizing a clear and reliable process of thinking with images.

## Abstract

Recent studies have demonstrated the efficacy of using Reinforcement Learning (RL) in building reasoning models that articulate chains of thoughts prior to producing final answers. However, despite ongoing advances that aim at enabling reasoning for vision-language tasks, existing open-source visual reasoning models typically generate reasoning content with pure natural language, lacking explicit integration of visual information. This limits their ability to produce clearly articulated and visually grounded reasoning chains. To this end, we propose Grounded Reasoning with Images and Texts (GRIT), a novel method for training MLLMs to think with images. GRIT introduces a grounded reasoning paradigm, in which models generate reasoning chains that interleave natural language and explicit bounding box coordinates. These coordinates point to regions of the input image that the model consults during its reasoning process. Additionally, GRIT is equipped with a reinforcement learning approach, GRPO for Grounded Reasoning (GRPO-GR), built upon the GRPO algorithm. GRPO-GR employs robust rewards focused on the final answer accuracy and format of the grounded reasoning output, which eliminates the need for data with reasoning chain annotations or explicit bounding box labels. As a result, GRIT achieves exceptional data efficiency, requiring as few as 20 image-question-answer triplets from existing datasets. Comprehensive evaluations demonstrate that GRIT effectively trains MLLMs to produce coherent and

visually grounded reasoning chains, showing a successful unification of reasoning and grounding abilities. All code, data, and checkpoints will be released.

# 1 Introduction

Reasoning models [1, 2, 3, 4, 5] are trained to articulate their problem-solving process through a "reasoning chain" which comprises a detailed, step-by-step deliberation and a final answer. Recent studies have shown that such trained reasoning models are superior in reasoning than models directly prompted in the zero-shot Chain-of-Thought (CoT) [6] manner, where off-the-shelf models are simply guided to "think aloud" at inference time, often reflecting little of their true internal processes [7]. Reasoning models first became particularly prominent in the language domain [2, 6, 5, 8, 9], with models like DeepSeek-R1 [2] showcasing capabilities for complex tasks, and subsequently extended to the vision-language field [10, 11, 12].

Current open-source vision reasoning models yield reasoning chains consisting only of natural language when faced with multimodal inputs. This results in reasoning that is less clear and insufficiently grounded in the details of the visual input. To overcome this, a crucial question is how to empower models to think with images, mirroring how humans refer to visual input in their thoughts. However, realizing this capability presents significant technical challenges. Current MLLMs are designed to generate language tokens, lacking the inherent mechanism to generate images directly within a reasoning chain. Furthermore, processing and understanding reasoning sequences that interleave multiple visual elements poses a substantial hurdle, as many MLLMs struggle with maintaining context across numerous images in one input. Beyond these technical challenges, data also presents a major hurdle. In most cases, there is no unique correct reasoning path for complex multimodal questions, and human-annotated data explicitly incorporating visual evidence in reasoning steps is severely scarce.

To address these challenges and enable more grounded visual reasoning in MLLMs, we propose the Grounded Reasoning with Images and Text (GRIT) method. GRIT introduces a novel grounded reasoning paradigm where the model generates natural language reasoning chains that freely mix bounding box coordinates pinpointing relevant regions from the input image. These bounding boxes serve to indicate the specific visual information that the model is consulting in its reasoning process. To simplify the inference process, after the generation of bounding box coordinates, the model does not receive additional pixel inputs in the proposed grounded reasoning paradigm; instead, the model comprehends and utilizes the visual information indicated by these coordinates based on its understanding of the original input image. By cropping the input image with the generated bounding boxes, the resulting reasoning chain can be visualized as interleaved text and highlighted regions from the input image, as illustrated in Figure 1.

To train MLLMs to produce reasoning chains in the grounded reasoning paradigm, GRIT includes a reinforcement learning method, GRPO for Grounded Reasoning (GRPO-GR), built upon the GRPO algorithm. It is equipped with novel rewards specifically focused on the format of not only reasoning but also grounding, in addition to answer accuracy. Specifically, such format reward encourages reasoning outputs structured by a thinking token pair (e.g., <think> and </think>) and a rethink token pair (e.g., <rethink> and </rethink>); it also rewards the inclusion of syntactically valid bounding boxes within the generated sequence. As a result, the rewards in GRPO-GR do not constrain the specific textual content of the reasoning steps or the semantic accuracy of the grounded regions, thus eliminating the need for data with reasoning chain annotations or explicit bounding box labels. As a result, we find that the GRIT method is extremely data efficient: it enables MLLMs to acquire the grounded reasoning ability with very few data samples sourced from existing VQA datasets using only image-query-answer triplets.

With the GRIT method, we train state-of-the-art MLLMs—Qwen 2.5-VL [3] and InternVL 3 [13] using only 20 image–question–answer triplets drawn from existing object-relation and counting VQA datasets, VSR [14] and TallyQA [15]. A significant outcome of GRIT is that the trained models preserve their broad versatility, effectively handling not only visual question answering but also grounding-heavy referring expression comprehension tasks. In our experiments with a variety of testing data collected from benchmarks for both VQA and referring expression comprehension, we reveal several key observations. Firstly, the trained models effectively unify the grounding and reasoning abilities—which were originally inherent but disconnected in the base MLLMs—within

their grounded reasoning output. Secondly, through both qualitative and quantitative analysis, we reveal a high correlation between the image regions referenced and the accompanying text in the reasoning chain produced by GRIT-trained models. Furthermore, we demonstrate that the generation of bounding boxes boosts the subsequent model reasoning to attend more effectively to the input visual information. Finally, we observe that as training data increases, models trained with GRIT show improved performance but it also reveals challenges for boosting generalizability. Our contributions are as follows:

- We propose Grounded Reasoning with Images and Text (GRIT), a novel method that teaches MLLMs to think with images through a grounded reasoning paradigm where models generate reasoning chains interleaving natural language with explicit bounding box coordinates.

- We develop GRPO-GR, a reinforcement learning algorithm, which employs novel rewards that enable the grounded reasoning ability of MLLMs efficiently, using only image-question-answer triplets without requiring dense reasoning chains or bounding box annotations.

- Through comprehensive evaluations, we demonstrate that MLLMs trained with GRIT, such as Qwen 2.5-VL and InternVL 3, successfully unify their grounding and reasoning abilities to produce accurate and coherent grounded reasoning.

## 2 Related Work

### 2.1 Reinforcement Learning for Vision-Language Reasoning

Recent studies have applied reinforcement learning (RL) with verifiable rewards to build visual reasoning models for visual question-answering tasks, extending approaches from language-only models, such as DeepSeek-R1 [2], to enhance Multimodal Large Language Models (MLLMs). For instance, R1-OneVision [12] and R1-V [16] focus on diagram reasoning and math problems, respectively, while Vision-R1 [11] emphasizes symbolic reasoning tasks. However, these methods often treat visual grounding and textual reasoning as separate or do not tightly integrate them into a single generative process. VLM-R1 [10] applies RL to referring expression comprehension tasks, a grounding-heavy task, rewarding bounding box, and answer accuracy. While effective for these tasks, VLM-R1 typically outputs only bounding boxes as final answers, with an implicit reasoning process, rather than an interpretable, interleaved trace of text and visual grounding. In contrast, our GRIT (Grounded Reasoning with Images and Text) framework uses RL to train MLLMs to freely mix grounding and reasoning within a single generative trace. Models trained with GRIT produce interleaved chains of natural language and bounding box coordinates, enabling a dynamic interplay where visual evidence informs textual logic, and vice-versa. While proprietary systems such as ChatGPT-o3/4 [17] have shown similar "thinking with images" capabilities, GRIT offers the first open-source approach to achieve this interleaved visual-textual reasoning via lightweight RL, without needing explicit annotations for intermediate reasoning or grounding steps.

### 2.2 Visual Chain-of-Thought Reasoning

The idea of Chain-of-Thought (CoT) reasoning for vision-language tasks predates the RL-focused methods [18, 19, 20, 21], where models are prompted to generate reasoning chains that include visual cues. Early approaches like Multimodal-CoT [19] used multi-stage prompting, while others like CCoT [21] leveraged external tools like scene graphs. These often rely on prompting or auxiliary modules rather than learning an end-to-end generative process for interleaved reasoning. Other works aimed to learn visually grounded CoT with minimal supervision. UV-CoT [22] used self-generated bounding boxes and an auxiliary MLLM for supervision, but still largely separated the grounding and reasoning phases. Pioneer works such as VisCoT [23], CogVLM [24] and CogCoM [25] fine-tune models on datasets with detailed annotations for both textual rationales and corresponding bounding boxes. However, this requires high-quality, dense annotations linking each reasoning step to specific visual evidence. The GRIT method differs by enabling MLLMs to generate explicit, interpretable, and visually grounded reasoning steps from task-level reward signals alone, without requiring supervisory signals for bounding boxes or intermediate textual thoughts within the reasoning chain. Visual grounding (via bounding box coordinates) is embedded within the continuous reasoning chain. This allows models trained with GRIT to achieve a form of grounded CoT where visual information is

directly integrated into the thought process, enabling them to reason "with" images, not just "about" them.

## 3 GRIT: Grounded Reasoning with Images and Text

### 3.1 Grounded Reasoning Paradigm

The GRIT framework fosters a straightforward, grounded reasoning paradigm in MLLMs, enabling a more transparent and verifiable reasoning process. Given an image $I$ and a textual question $q$, GRIT enables the model to generate a two-part output $(c, a)$: first, a reasoning chain $c$ (starting with `<think>`), followed by a concise final answer $a$ (after `<answer>`). The reasoning chain $c$ freely mixes natural-language text $T$ and optional bounding-box coordinates $B$. At any step $p$ of the token generation of $c$, the model can choose to generate a bounding box $c_p \in B$ or it can opt to continue generating natural language text $c_p \in T$. The decision is based on the input and all existing reasoning chain $c_{1:p-1}$. When the model has finished generating one or multiple bounding box coordinates at step $q$, these coordinates are intended to directly inform and shape the subsequent reasoning steps $c_{q+1}, c_{q+2}, \dots$. Crucially, the generation of subsequent tokens does not receive additional pixel inputs based on the generated bounding boxes. Instead, the model relies on its internal understanding, informed by these newly generated coordinates, to continue the reasoning process. This requires the model to learn to interpret its own grounding actions i.e. the bounding boxes, and integrate that understanding into its ongoing textual deliberation. Leveraging the inherent grounding and reasoning abilities in MLLMs, the proposed grounded reasoning paradigm encourages the model to unify these existing faculties to form the new grounded reasoning ability. Compared to alternative approaches such as generating pixel-level attention masks or adding the image region indicated by bounding boxes as additional input for multi-turn generation, the grounded reasoning paradigm is significantly more efficient.

### 3.2 Reinforcement Learning with GRPO-GR

The GRIT method trains MLLMs via a newly proposed reinforcement learning algorithm, GRPO for Grounded Reasoning (GRPO-GR), for grounded reasoning ability. Built upon the Group-Relative Policy Optimisation (GRPO) [26] algorithm, GRPO-GR optimizes a policy $\pi_\theta$ to generate sequences of reasoning $(c, a)$ based on rewards combining answer correctness with format adherence as shown in Figure 2. A fixed prompt suffix is appended to the model's input during training and inference, please refer to the Appendix D for details.

**RL Formulation.** The model acts as a policy $\pi_\theta$ that generates the output sequence $(c, a)$ given the input $(I, q)$. During training, for every image–question pair $(I, q)$, we sample a group of $N$ candidate completions $\{o_1, \dots, o_N\}$ from the current policy $\pi_\theta$. For each completion $o_i$, a task reward $r_i = R(q, o_i)$ is computed based on a combination of components (detailed below). These rewards are used to derive a group-normalised advantage:

$$A_i \;=\; \frac{r_i - \mathrm{mean}\{r_1, \dots, r_N\}}{\mathrm{std}\{r_1, \dots, r_N\} + \delta}, \tag{1}$$

where $\delta$ is a small constant (e.g., $10^{-8}$) for numerical stability.

The task reward $r_i$ is a composite signal comprising three components: a grounded-reasoning-format reward ($r_{\mathrm{format}}$), an optional grounded-target-counting reward ($r_{\mathrm{count}}$), and a GPT-aided answer-accuracy reward ($r_{\mathrm{ans}}$). These components are designed to encourage the desired grounded reasoning behavior and accurate final answers.

**Grounded-reasoning-format reward ($r_{\mathbf{format}}$).** This reward encourages reasoning outputs structured by special token pairs and includes syntactically valid bounding boxes. It is composed of a special-token-format signal $s_{\mathrm{st}}$ and a bounding-box-format signal $s_{\mathrm{bf}}$:

$$r_{\mathrm{format}} = s_{\mathrm{st}} + s_{\mathrm{bf}}, \tag{2}$$

where $s_{\mathrm{st}}$ rewards the correct usage and order of special reasoning-format tokens (`<think>`...`</think>` then `<rethink>`...`</rethink>`) within the reasoning chain $c$, which structure the reasoning, potentially across multiple steps. Each correctly placed token pair increments the reward by 0.5: $s_{\mathrm{st}} = 0.5 \times \mathbb{I}(\text{correct think token pair}) + 0.5 \times \mathbb{I}(\text{correct rethink token pair})$. The

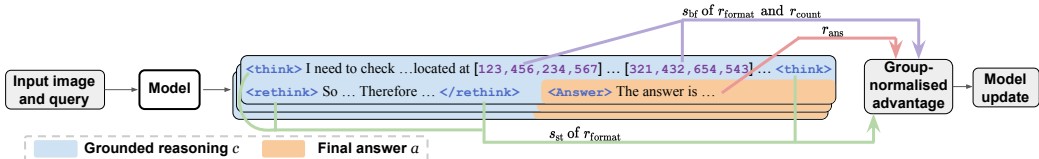

Figure 2: Model update via GRPO-GR. During GRPO-GR training, we sample a group of model completions and calculate the grounded-reasoning-format reward ($r_{\text{format}}$), the optional grounded-target-counting reward ($r_{\text{count}}$), and the GPT-aided answer-accuracy reward ($r_{\text{ans}}$). The rewards are used to calculate the group-normalized advantage and guide the policy optimization.

bounding box format signal $s_{\text{bf}}$ incentivizes the explicit generation of syntactically correct bounding boxes within $c$. These are detected via a regex matching quadruplets of integers separated by commas, typically expected before a rethink token. A reward of $0.5$ is assigned if at least one such bounding box is present: $s_{\text{bf}} = 0.5 \times \mathbb{I}(\text{num\_bboxes} \geq 1)$. This reward component encourages the required format and presence of visual grounding elements without constraining the textual content or semantic accuracy of the grounded regions themselves.

**Grounded-target-counting reward ($r_{\text{count}}$).** This optional reward component is used specifically for training examples drawn from visual counting-related datasets (detailed in the experimental setup). It is set to $0.5$ if the *number* of bounding boxes generated within the reasoning exactly matches the ground-truth count for the target object. This encourages the model to systematically generate the correct quantity of bounding boxes as part of its counting reasoning process.

**GPT-aided answer-accuracy reward ($r_{\text{ans}}$).** This reward combines signals for the correctness of the final answer, offering a more robust signal than rule-based checks alone by supplementing them with an external Vision-Language Model judge. It is computed as: $r_{ans} = s_{GPT} + 0.1\, s_{BLEU}$, where $s_{GPT}$ is a binary correctness score (0 or 1) from a GPT-4o judge evaluating the question, predicted answer, and ground truth triplet $(q, \hat{a}, a)$, and $s_{BLEU}$ is the sentence-level BLEU-1 similarity between $\hat{a}$ and $a$. The GPT-4o prompt is fixed and detailed in Appendix D. We down-weight $s_{BLEU}$ due to its sensitivity to length mismatch, ensuring the highest reward is for closely matching ground truth answers.

**Optimization Objective.** The policy parameters $\theta$ are updated by maximizing the GRPO objective $\mathcal{J}_{\text{GRPO}}(\theta)$ using the calculated group-normalised advantages $A_i$:

$$\mathcal{J}_{\text{GRPO}}(\theta) = \frac{1}{N} \sum_{i=1}^{N} \Big[ \min\big(s_i A_i,\ \text{clip}\big(s_i,\ 1-\epsilon,\ 1+\epsilon\big) A_i\big) - \beta\, D_{KL}\big(\pi_\theta(\cdot|q) \,\|\, \pi_{\text{ref}}(\cdot|q)\big) \Big], \quad (3)$$

where $\pi_{\text{ref}}$ is the reference policy (typically the initial pre-trained model), $\beta$ controls the KL divergence penalty, $\epsilon$ defines the trust region for the importance ratio $s_i = \frac{\pi_\theta(o_i|q)}{\pi_{\theta_{\text{old}}}(o_i|q)}$, and $\theta_{\text{old}}$ are the policy parameters before the update. This objective balances maximizing expected reward within each group against maintaining proximity to the reference policy, promoting stable learning.

## 4 Experiments

We first evaluate the grounded reasoning performance of models trained using the GRIT method in both grounding and reasoning perspectives. Then, we further qualitatively and quantitatively analyze the interaction between the bounding boxes and other reasoning contents generated during grounded reasoning. Last but not least, we examine the impact of increasing the training data volume.

### 4.1 Setup

**Testing data.** We evaluate models trained with GRIT on curated testing sets derived by sampling from six public datasets: Visual Spatial Reasoning (VSR) [14] focusing on spatial relation verification, TallyQA [15] on object counting, GQA [27] on compositional object spatial questions, MME [28] on diverse visual tasks including counting and position, MathVista-mini [29] on mathematical reasoning in visual contexts, and position subset of OVDEval [30] on open-vocabulary object grounding. For assessing the quality of bounding boxes generated during grounded reasoning outputs, we leverage

Table 1: Evaluation of the grounded reasoning accuracy. GRIT-trained models are compared with baselines across seven testing sets on GPT-as-judge answer accuracy score (ACC) and grounding IoU (GIoU). GRIT-trained models overall outperform baselines, demonstrating a successful unification of grounding and reasoning abilities that are originally inherent but separated in MLLMs.

| | VSR | | TallyQA | | GQA | | MathVista | MME | OVDEval |
|---|---|---|---|---|---|---|---|---|---|
| | ACC | GIoU | ACC | GIoU | ACC | GIoU | ACC | ACC | GIoU |
| Qwen2.5-VL 3B [8] | | | | | | | | | |
| Direct query | 49.5 | 0.00 | 40.8 | 0.00 | 55.4 | 0.00 | 58.5 | 88.9 | 0.389 |
| Chain-of-Thought | 37.5 | 0.122 | 33.2 | 0.113 | 39.5 | 0.269 | 33.0 | 41.3 | 0.388 |
| One-shot ICL | 13.2 | 0.213 | 36.3 | 0.268 | 20.4 | 0.441 | 29.1 | 24.7 | 0.328 |
| Few-shot fine-tuning | 59.7 | 0.216 | 44.5 | 0.284 | **64.6** | 0.475 | 45.0 | 68.3 | 0.391 |
| GRIT | **72.9** | **0.325** | **47.8** | **0.447** | 62.8 | **0.485** | **59.8** | **89.3** | **0.398** |
| InternVL3 2B [13] | | | | | | | | | |
| Direct query | 52.9 | 0.000 | 15.5 | 0.000 | 29.4 | 0.000 | 43.0 | 40.0 | 55.1 |
| Chain-of-Thought | 6.4 | 0.428 | 6.8 | 0.279 | 4.1 | 0.292 | 7.5 | 14.0 | 21.9 |
| One-shot ICL | 3.4 | 0.435 | 3.7 | 0.275 | 2.6 | 0.435 | 14.1 | 1.3 | **59.3** |
| Few-shot fine-tuning | 54.0 | 0.381 | 22.5 | 0.116 | 46.8 | 0.114 | 17.4 | 62.3 | 7.8 |
| GRIT | **64.9** | **0.495** | **44.2** | **0.324** | **63.2** | **0.457** | **48.2** | **82.0** | 56.0 |

available question-related bounding box annotations from VSR, TallyQA, and GQA and manually refine the bounding box coordinates for VSR and GQA data for our evaluation purpose. We provide more details, including data statistics, in Appendix A.

**Training Data.** Demonstrating the data efficiency of our GRIT method, we train on a dataset of only 20 unique image-query-answer triplets. This small training set is drawn from the Visual-Spatial Reasoning (VSR) [14] and TallyQA [15] datasets. These data focus on tasks requiring both explicit visual grounding and multi-step reasoning, providing a suitable testbed to evaluate GRIT's ability to learn grounded reasoning formats with limited data. Please refer to Appendix B for more details.

**Training Implementation.** We train two pre-trained MLLMs, Qwen2.5-VL-3B and InternVL-3-2B, directly using the GRIT method with reinforcement learning. We train the models for 200 steps with a total batch size of 128. During GRPO-GR training, we generate 4 candidate reasoning traces per input sample during training with a learning rate of 2×10 e-6. The optimizer for the training is AdamW and a Cosine scheduler is adopted. All training is conducted on 8 NVIDIA A100 (80GB) GPUs with Deepspeed Zero2 and the time for training each model is approximately 12 hours.

### 4.2 *Research question 1*: How accurate is the grounded reasoning from both grounding and reasoning perspectives?

**Metrics.** To comprehensively assess model performance across testing sets, we evaluate two key aspects of their output: (1) the correctness of the natural language answer, evaluated by *GPT-as-judge answer accuracy score* [31], a score between 0 (completely incorrect) and 1 (fully correct) assigned by GPT-4o to judge the correctness of the model-generated natural language answer. The GPT-4o is provided with the same prompt format as in the GPT-aided answer-accuracy reward during training (Section 3.2); (2) the accuracy of the grounding coordinates, measured by *grounding IoU*, the average Intersection over Union (IoU) between the union of all bounding boxes generated by the model (i.e., within its reasoning chain or as the direct answer for tasks like OVDEval) and the union of all corresponding ground-truth bounding boxes. Different from metrics in traditional object detection tasks, grounding IoU focuses on whether the collection of grounded regions in each grounded reasoning output from the model together aligns with the annotated question-critical image regions.

**Baselines.** We include baselines built upon the same base MLLMs and the same data as the models trained with GRIT: (1) *Direct Query* directly feeds the original MLLMs only the task query alongside the input image in a zero-shot manner, without any specific formatting or reasoning prompts, to assess its raw task-solving ability. (2) *Chain-of-Thought (CoT)* [6] feeds the original MLLMs queries with a fixed prompt suffix, which instructs the model to generate step-by-step reasoning chains before the answer. The prompt suffix for the CoT baseline is identical to the prompt used for models trained with GRIT, measuring the model's latent ability without any post-training. (3) *One-shot In-Context*

*Learning (ICL)* [32] prefixes each test question with a single, fixed exemplar consisting of {question, grounded-visual-thinking trace, answer}, instructing the model to imitate the format and reasoning style. (4) *Few-shot SFT* fine-tunes the MLLM on the same demonstrations employed for GRPO but with standard supervised learning, isolating the effect of reinforcement learning.

**Results.** The results are summarized in Table 1. In contrast to baselines, models trained with GRIT overall achieve higher scores on the adopted testing data. Despite being trained on only 20 training samples, models trained with GRIT not only improve on GPT-as-judge answer accuracy Scores on VSR and TallyQA (the two datasets seen during training) but also generalize effectively to other out-of-domain data, indicating strong reasoning ability. The results on the grounding IoU metric show that models trained with GRIT, although not directly optimized for this metric, outperform baselines in locating question-relevant image regions during their reasoning. Notably, on the OVDEval testing data, models trained with GRIT achieve more accurate detection results than zero-shot MLLMs, highlighting their emerging improvements in grounding capabilities. These results demonstrate a more successful unification of grounding and reasoning, where their integration demonstrably enhances the performance of both individual abilities.

From the result, we also observe that baselines based on off-the-shelf MLLMs exhibit rigidity, where they tend to generate either only bounding boxes or the final answer. For CoT and one-shot ICL baselines, although they are prompted or guided by ICL to produce a reasoning chain interleaved with bounding box coordinates, their grounding and reasoning functions are forced to operate concurrently. As a result, they generally face a severely deteriorated performance either in answer accuracy or grounding IoU, indicating that these capabilities remain largely separated and can interfere with each other, resulting in suboptimal performance. This suggests a general disconnect between their inherent grounding and reasoning abilities. As for the Few-shot SFT baseline, while demonstrating more balanced performance across grounding IoU and GPT-as-judge answer accuracy score compared to the zero-shot baselines, it still achieves considerably lower scores than models trained with our GRIT method. This is likely because supervised fine-tuning primarily learns to mimic the surface form of outputting text interspersed with bounding boxes, rather than developing a deeply integrated reasoning process where grounding logically informs each step.

### 4.3 *Research question 2*: How generated bounding boxes interact with natural-language contents during grounded reasoning?

**Qualitative Analysis of Grounded Reasoning Contents.** We qualitatively showcase the grounded reasoning produced by models trained with GRIT through a variety of input-output examples. A major strength of training with the GRIT method is that its reward design imposes no rigid constraint on the content of the reasoning chain. Consequently, the model learns flexible reasoning strategies that adapt seamlessly to diverse queries. Specifically, as shown in Figure 3i, GRIT-trained models sometimes directly provide an answer while then reflecting the answer in subsequent reasoning; while in other instances, as in Figure 3ii, they perform the ground actions to identify visual evidence initially and subsequently analyze the selected image regions in their reasoning. Crucially, our models dynamically determine whether grounding is necessary, significantly reducing false-positive grounding instances. For example, in Figure 3iii, queries referencing non-existent entities in the input image do not prompt erroneous groundings, showcasing a robust multimodal reasoning capability.

**Cross-modal Correlation of Images Regions and Thoughts.** To systematically evaluate the coherence between image regions and the natural language contents interleaved in the grounded reasoning chain of models trained with GRIT, we introduce the *Vision-Language Reasoning Cross-Modal Correlation* metric. Given each model-generated reasoning chain ($c$), we extract the associated bounding boxes $\{c_i | c_i \in B\}$. To establish a rigorous evaluation, we randomly sample an equal number of bounding boxes from the input image as negative candidates $\{h_0, ... h_j\} \in B$. We draw these two sets of boxes separately on the input image and then ask GPT-4o to identify the one from two input images with the set of bounding boxes that corresponds most closely with the textual reasoning with bounding box coordinates masked. Leveraging GPT-4o's strong Set-of-Mark (SoM) capabilities [33], this process allows us to quantitatively assess semantic coherence. We repeat this process three times for robustness and report the average correlation score with standard deviation. We evaluate the models trained with GRIT, Zero-shot ICL, and Few-shot SFT outputs using this cross-modal correlation metric on six testing sets (VSR, TallyQA, GQA, MME, and MathVista), excluding OVDEval as it primarily poses a grounding challenge. In addition to evaluating model-generated

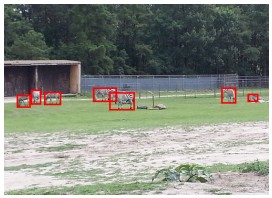

**Question**: How many zebras are pictured here?
**Model output**:
<think> There are six zebras in the picture. The coordinates for the zebras are as follows: 1. (200, 168, 248, 202) 2. (169, 159, 214, 186) 3. (76, 167, 108, 192) 4. (24, 173, 50, 197) 5. (51, 163, 70, 191) 6. (413, 159, 441, 189) 7. (463, 171, 483, 186) </think>
<rethink> The coordinates provided for the zebras are accurate and cover all the zebras visible in the image. There are no overlapping or missing coordinates. </rethink>
<answer> 7
**Ground truth answer**: 7

(i) Model provides an answer with regions and then reflects on it in the subsequent reasoning.

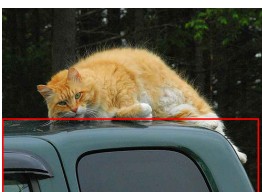

**Question**: Is the truck beneath the cat?
**Model output**:
<think> The bounding box coordinates for the truck are approximately (0, 209, 488, 364). The cat is positioned on top of this truck. </think>
<rethink> The cat is on top of the truck, so the truck is beneath the cat. </rethink>
<answer> Yes
**Ground truth answer**: Yes

(ii) Model first grounds critical image region in its reasoning and then analyze in the subsequent reasoning.

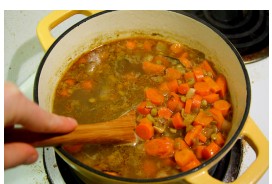

**Question**: Is there a knife in the image?
**Model output**:
<think> There is no knife present in the image. The focus is on a pot of soup with carrots and other ingredients. The bounding box coordinates for the knife would be outside the image area. </think>
<rethink> The image shows a pot of soup with carrots and other ingredients, but there is no knife present. The bounding box coordinates for the knife would be outside the image area. </rethink>
<answer> No
**Ground truth answer**: no

(iii) Model correctly handles queries about non-existent entities without any grounding action.

Figure 3: Inference examples of Qwen2.5-VL-GRIT.

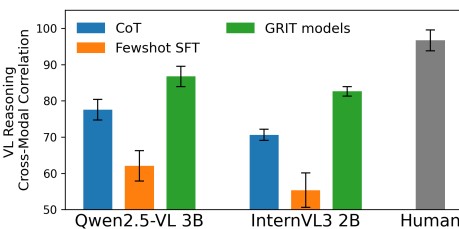

Figure 4: Correlation between image regions and "thoughts" in grounded reasoning evaluated by our Vision-Language Reasoning Cross-Modal Correlation metric. The result shows that models trained with GRIT outperform baselines.

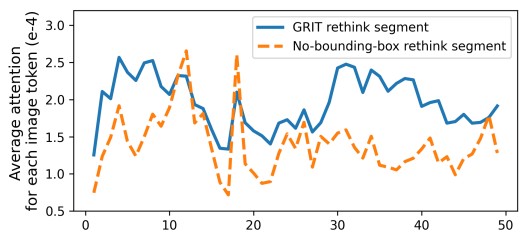

Figure 5: Model's average attention for image tokens during the generation of rethink segments. The overall higher curve for the original rethink segments of the GRIT-trained model shows that the bounding boxes generated facilitate stronger attention to the image input in subsequent reasoning.

outputs, we also manually create and evaluate 20 human-written reasoning chains with interleaved bounding boxes using the same vision-language reasoning cross-modal correlation method to establish a human performance baseline. As shown in Figure 4, models trained with our GRIT framework outperform both Zero-shot ICL and Few-shot SFT, showing highly correlated image region selection with textural reasoning, while still exhibiting a gap when compared to the human-written reasoning chains, indicating room for future improvement.

**Influence of Grounding on Subsequent Reasoning.** Models trained with GRIT interleave reasoning with grounding actions when producing visually integrated thought processes. To further understand how the bounding boxes generated impact subsequent reasoning, we examine attention scores for

input visual tokens during inference. Specifically, we split outputs of a Qwen2.5-VL model trained with GRIT using the `<rethink>` token into pre-rethink and rethink segments, where pre-rethink segments usually include bounding-box coordinates due to the optimization guided by the grounded-reasoning-format reward detailed in Section 3.2. We then create an alternative pre-rethink segment by removing all bounding boxes from the pre-rethink segment, simulating the situation where no grounding action is done during the reasoning. Next, we feed the alternative pre-rethink segment back into the Qwen2.5-VL model trained with GRIT for continuous token generation. We refer to the newly generated content based on the modified input as the no-bounding-box rethink content. Finally, we compute and compare average attention scores across various layers for input visual tokens during the generation of both the original and no-bounding-box rethink content. Such comparison is repeated for 100 randomly selected data samples in the GQA subset, and due to the various generation lengths, we show the results for the first 50 tokens generated. From the results shown in Figure 5, we find that the average attention scores to each visual token is overall significantly higher in the original rethinking segment than the no-bounding-box rethink segment. This indicates the presence of bounding boxes in the original pre-rethink segment leads to increased visual attention in the following reasoning chain, which potentially benefits the image consistency of the reasoning process.

### 4.4 *Research question 3*: What is the effect of scaling training data?

To understand how GRIT's performance scales with increasing training data, beyond the data efficiency shown with 20 samples, we trained variants of the Qwen2.5-VL 3B model using 20, 500, and 7,000 image–question–answer triplets. These samples were taken from the VSR and TallyQA datasets, maintaining the same mix of data sources. We evaluate these trained variants on the testing data sets introduced in Section 4.1 with the GPT-as-judge answer accuracy score. Figure 6 presents the results across testing data categorized as either in-domain, sourced from the same datasets as training (VSR, TallyQA), or out-of-domain (GQA, MathVista-mini). The results show that answer accuracy generally increases with more training data used for GRIT. We find that the improvements in answer accuracy are more subtle for out-of-domain testing data compared to the growth

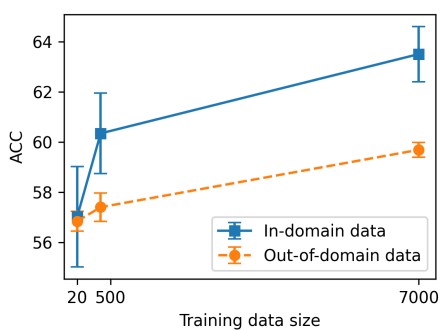

Figure 6: Effect of scaling training data on model performance.

observed on in-domain testing data, which highlights a common challenge of generalization. Furthermore, we observe that the model's performance growth diminishes, suggesting that continued improvement may require exponentially more data. We interpret this observation in light of recent studies suggesting that RL with verifiable rewards primarily biases existing reasoning patterns towards rewarded outcomes rather than fundamentally changing them [34], with performance also heavily influenced by pre-training [35, 36]. This implies that to significantly enhance the out-of-domain performance of GRIT-trained models, increasing the diversity of training data is more critical than simply scaling up the data volume, highlighting a key direction for future study.

## 5   Conclusion

In this paper, we propose Grounded Reasoning with Images and Text (GRIT), a novel method where models generate visually grounded reasoning chains by interleaving natural language with explicit bounding box coordinates referencing relevant image regions. We propose the grounded reasoning paradigm and the GRPO-GR reinforcement learning training method. As a result, GRIT is extremely data efficient and requires neither dense reasoning chains nor explicit bounding box annotations. Our experiments demonstrate that GRIT effectively trains pre-trained MLLMs to produce accurate and coherent grounded reasoning. While limited resources focus our study on smaller MLLMs to validate the GRIT method and efficiency, rather than exploring peak performance with full-scale scaling, we identify key direction for further improving the generalizability of GRIT-trained model being data variety and model pre-training. Nonetheless, GRIT's efficacy in instilling complex grounded reasoning with minimal data highlights its significant potential.

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

# A    Details for Testing Data

For our evaluation, we curated testing data derived from six public open-source datasets covering a range of visual reasoning and grounding tasks. The statistic for the testing data is shown in Table 2.

- *VSR* [14] tests spatial relation verification. For our VSR evaluation set, we source question-image-answer triplets from the VSR subset of the Visual CoT benchmark [23] and manually filter out those with ambiguous answers.

- *TallyQA* [15] focuses on counting; we uniformly sample evaluation questions where the target object counts range from 0 to 9 to create our TallyQA evaluation set.

- *GQA* [27] offers scene-graph-grounded, compositional object spatial questions. We first take the GQA subset from the Visual CoT benchmark and then manually filter these to retain high-quality instances for our GQA evaluation set.

- From *MME* [28], we use only the counting, position, and existence subsets to broaden our evaluation scope.

- *MathVista* [29] evaluates mathematical reasoning in visual contexts. Following prior works, we adopt its `TestMini` split.

- Finally, *OVDEval* [30] is an open-vocabulary detection (OVD) testing set that requires the model to ground fine-grained semantics from the language query to the coordinates of visual features. We use its position subset and simplify it to object detection tasks with a single target.

Among these evaluation sets, those derived from VSR, TallyQA, and GQA are accompanied not only by ground-truth language answers but also by annotations of bounding boxes for image regions critical for deriving the answer. Specifically, we manually refine the bounding box coordinates from the Visual CoT benchmark for our VSR and GQA data. For our TallyQA evaluation sets, we adopt the original bounding box annotations. All evaluation sets, except for OVDEval, are VQA benchmarks where the required answer is a single word or short phrase. In contrast, OVDEval differs as grounding is not an optional component of the reasoning chain but is explicitly required as the answer to queries.

# B    Details for Training data

To demonstrate the data efficiency of our GRIT method, we collect a small training dataset consisting of only 20 unique image-query-answer triplets. These triplets are sourced from existing open-source VQA datasets, covering both grounding and visual reasoning challenges. Specifically, we selected ten from the Visual Spatial Reasoning (VSR) dataset [14] and ten from the TallyQA dataset [15]. We chose examples from VSR as they typically involve object localization and spatial relation reasoning (e.g., "Is the motorcycle away from the bird?"). From TallyQA, we selected counting tasks (e.g., "How many signs are on the post?"), specifically ensuring the chosen examples uniformly cover object counts from 0 to 4, as these tasks naturally lend themselves to explicit grounded counting within the reasoning process. This curated dataset engages the models in multi-step visual analysis, serving to evaluate GRIT's ability to strengthen the link between visual grounding and logical deliberation.

To validate the importance of including counting-related training data and the associated grounded-target-counting reward within the GRIT method, we conduct an ablation experiment. The standard training utilizes data from TallyQA, which consists of queries about object quantities in images with single-digit numerical answers. For these counting tasks, we employ the grounded-target-counting reward (detailed in Section 3.2), designed to encourage the generation of a flexible number of bounding boxes matching the count in the reasoning output. In the ablation, we train the InternVL-3 2B model with a modified dataset and reward function. Instead of using 10 VSR and 10 TallyQA samples with the full reward set, we train with a dataset of 20 VSR samples and exclude the grounded-target-counting reward during training. This allows us to isolate the contribution of the counting-related data and reward component. We report the GPT-as-judge answer accuracy score and the grounding IoU of both in-domain and out-of-domain data (same as in Section 4.4). The results, presented in Table 3, show that excluding the counting-related data and grounded-target-counting reward during training leads to a significant performance decrease in grounding, as indicated by a lower Grounding IoU score compared to the original GRIT-trained model. Furthermore, we observe

Table 2: Statistics for the testing data used in the experiments. We collect the testing data from six diverse benchmarks.

| Data source | VSR | TallyQA | GQA | MathVista | MME | OVDEval |
|---|---|---|---|---|---|---|
| Counts | 288 | 491 | 509 | 1000 | 240 | 2164 |
| Avg question/answer length | 6.7/1.0 | 6.0/1.0 | 7.1/1.0 | 38.2/1.2 | 13.3/1.0 | 16.4/4 |
| Ratio of multi-choice and yes/no questions (%) | 71.2 | 0 | 58.9 | 70.8 | 100 | 0 |
| Ratio of annotated grounding targets (%) | 58.8 | 25.6 | 25.3 | - | - | 17.3 |

Table 3: Ablation study on the importance of counting data and grounded-target-counting reward. Comparison of the original GRIT-trained model, trained with 10 VSR + 10 TallyQA and counting reward, with an ablated variant, trained with 20 VSR without counting reward. Results show a performance decrease in the ablated model.

| | GIoU | | ACC | |
|---|---|---|---|---|
| | In-domain | Out-of-domain | In-domain | Out-of-domain |
| GRIT | **0.387** | **0.437** | 51.8 | **64.4** |
| GRIT w/o counting data & reward | 0.349 | 0.378 | **53.8** | 60.0 |

that this exclusion negatively impacts the model's answer accuracy on out-of-domain data. As a result, this ablation study underscores the importance of including both counting-related training data and the grounded-target-counting reward within the GRIT method.

## C  Ablation on Counting-related Training Data and Reward

## D  Prompts

We append the prompt shown in Figure 7 to the GRIT model training and model inference, as well as for the Chain-of-Though baseline in the experiments. The prompt provides models with the instruction to follow the grounded reasoning paradigm, however, as shown in the experiment result, MLLMs without training will face a significant performance drop. This is due to the instruction in the prompt requiring the MLLMs to perform grounding and reasoning at the same time, which is very challenging for them in a zero-shot manner. It is worth noticing that although the prompt mentions using "JSON" to show bounding boxes, the GRIT-trained model does not always adhere to such a format. This is intentional, as the reward in GRPO-GR uses regex to identify valid bounding boxes, rather than relying on the JSON format, providing the RL optimization of model policy with more search space. It also indicates that the prompt suffix only provides an initialization for the policy which can be adjusted with minimal impact as long as it includes an instruction of the grounded reasoning paradigm.

> First, think between <think> and </think> while output necessary coordinates needed to answer the question in JSON with key 'bbox_2d'. Then, based on the thinking contents and coordinates, rethink between <rethink> </rethink> and then answer the question after <answer>.

Figure 7: Prompt suffix that is appended to the input of models during the training and inference.

Additionally, in GPT-aided answer-accuracy reward of GRPO-GR and the GPT-as-judge answer accuracy score in the experiments, we adopt the prompt format in Figure 8.

Furthermore, in the experiments where we compute the Vision-Language Reasoning Cross-Modal Correlation metric for model outputs, we leverage GPT-4o to determine if the bounding boxes in the grounded reasoning chains of models are more correlated to the natural language reasoning contents than the randomly sampled bounding boxes. The prompt format we used is in Figure 9.

Figure 8: Prompt format for GPT-as-judge answer accuracy score and GPT-aided answer-accuracy reward. The $question and $answer are substituted with the query and ground truth in the data sample and $predicted_content is replaced by the model output.

Please decide which image has the bounding boxes that match the following description: {$grounded_reasoning_masked}
Reply with exactly "Image 0" or "Image 1".

Figure 9: Prompt format for calculating the Vision-Language Reasoning Cross-Modal Correlation metric. Here, the input images for GPT-4o are the same image drawn with two sets of bounding boxes, one from the generated grounded reasoning chain and one from random sampling. The $grounded_reasoning_masked is the generated grounded reasoning chain with any bounding box coordinates removed.

# E    More Examples

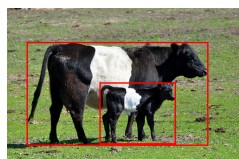

**Question**: What kind of animal stands on the ground?
**Model output**: <think> The image shows two animals standing on the ground. The larger animal is a cow with a white and black coat, and the smaller animal is a calf. The bounding box coordinates for the cow are approximately (42, 73, 433, 296), and for the calf, they are approximately (199, 160, 363, 293). </think> <rethink> The question asks about the animal that stands on the ground. Both the cow and the calf are standing on the ground, so either could be the correct answer. However, the cow is the larger and more prominent animal in the image. </rethink> <answer> cow
**Ground truth answer**: cow

(i) Our GRIT model accurately perceives the grounded target, reasons about the potential ambiguity in the question, and finally outputs a correct answer.

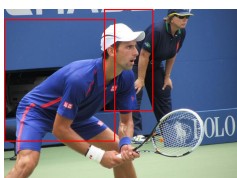

**Question**: Is there only one person appear in the image? Please answer yes or no.
**Model output**: <think> There is only one person in the image. bbox_2d_1: [0, 28, 305, 364] bbox_2d_2: [275, 0, 410, 280] </think> <rethink> The image shows two people, one in the foreground playing tennis and another in the background. bbox_2d_1: [0, 28, 305, 364] bbox_2d_2: [275, 0, 410, 280] </rethink> <answer> No
**Ground truth answer**: No

(ii) Although the bounding boxes generated during the grounded reasoning are slightly off, our GRIT model is still able to analyze the related image regions and successfully correct its own wrong answer at the beginning and finally generate a correct answer.

