# OpenReview forum: "GRIT: Teaching MLLMs to Think with Images"
_NeurIPS.cc/2025/Conference — NeurIPS 2025 poster_

### Official Review · Reviewer_zwCQ · 2025-07-02

**Clarity:** 4
**Significance:** 3
**Originality:** 3
**Rating:** 5
**Confidence:** 4

**Summary:**

This paper introduces GRIT, a method to train Multimodal Large Language Models (MLLMs) to generate reasoning chains that interleave natural language with explicit bounding box coordinates from input images. By combining reinforcement learning (via GRPO-GR) with task-specific rewards focused on answer accuracy, format correctness, and grounding validity, GRIT enables MLLMs to produce interpretable, visually grounded reasoning without requiring dense annotations. The method achieves strong performance on diverse vision-language tasks (e.g., counting, spatial reasoning, mathematical reasoning) using only 20 training samples. Experiments on models like Qwen2.5-VL and InternVL3 demonstrate improved QA and grounding accuracy.

**Questions:**

1. The training was conducted using only 20 samples, but it involved 200 training steps with a total batch size of 128, resulting in a computational cost of approximately 96 A100 hours. If we scale this setup linearly, would training with 7,000 samples consume as much as 33,600 A100 hours?
2. Furthermore, what would be the impact on model performance if the batch size or the number of training steps were reduced?
3. The data scaling experiments were conducted using `Qwen2.5-VL 2B` (Line 326), which does not appear to exist. Is this a typo?

**Ethical Concerns:**

["NO or VERY MINOR ethics concerns only"]

**Final Justification:**

The rebuttal has addressed all of my concerns. As a result, I decide to raise my rating and recommend acceptance of this paper.

**Limitations:**

Yes

**Quality:**

3

**Strengths And Weaknesses:**

### Strengths
1. GRIT introduces a paradigm shift by explicitly integrating visual grounding into reasoning chains, enabling MLLMs to "think with images" rather than just "about images." This bridges the gap between textual reasoning and visual evidence.
2. The method requires only 20 training samples, showcasing exceptional efficiency compared to existing approaches that often rely on large annotated datasets.
3. The authors validate their approach across seven datasets and provide both qualitative examples and quantitative analyses. Results show consistent improvements over baselines in answer accuracy and grounding precision.
4. The generated reasoning chains, with interleaved bounding boxes and text, offer transparency into the model's decision-making process, which is critical for trust and debugging.

### Weaknesses
1. While GRIT demonstrates success on smaller MLLMs (Qwen2.5-VL-3B, InternVL3-2B), the paper does not evaluate its scalability to larger models or more complex architectures, leaving questions about its applicability to state-of-the-art systems.
2. Experiments show diminishing returns with larger training datasets (e.g., 7,000 samples), suggesting that further improvements may require more diverse data. This highlights a potential bottleneck in real-world deployment.

---

> ### Author Rebuttal · Authors · 2025-07-30
>
> **Response to scalability to larger models**
>
> We appreciate the concern. This work aims to show that GRIT efficiently teaches pretrained MLLMs a unified grounding‑and‑reasoning ability with minimal training data. We therefore used smaller backbones to validate the method—especially its training‑data efficiency—and to isolate the effect of the grounded‑reasoning paradigm. Because smaller models are generally less capable than larger ones, we expect the observed gains to transfer on larger backbones. GRIT is model‑agnostic, specifying an interleaved text+box format and an RL objective without changing network internals, so it can be applied directly to larger models. A broader large‑model evaluation is planned for follow‑up work when additional computing resources are available, and we appreciate the reviewer’s understanding regarding typical academic compute budgets. We believe a proof‑of‑concept on compact models is crucial evidence that the paradigm itself drives the grounded‑reasoning gains.
>
>
>
>
> **Response to adequacy of accuracy for practical deployment**
>
> We thank the reviewer for raising this concern. Fig. 6 shows a sub‑linear performance improvements of the model with more data—an expected pattern in post‑training, however, the 7k‑sample result only reflects results under our intentionally constrained setting (eg. 3B backbone model, no domain‑specific warm‑start). It mainly underscores that besides the training data’s quantity, the in‑domain relevance plays an important role in model performance improvement. Real‑world deployment entails greater robustness requirements and design choices such as model size and cold‑start initialization; we leave a full deployment study to future work.
>
>
>
> **Response to question 1 on compute scaling with data size**
>
> We thank the reviewer for the detailed question. No—the cost does not scale linearly with data. Our 200 training steps are fixed across data sizes (RL update steps, not epochs). This is because we observe that with either 20 samples or 7k samples, performance saturates by ~150–200 steps. Although the 20‑sample training cycles through each data many more times than the 7k‑sample case (each example only ~3 effective epochs, ≈ 200×128/7,000), the GPU‑hours stay the same. We will further clarify this in our next paper revision.
>
> **Response to question 2 on reducing batch size or training steps**
>
> We thank the reviewer for the thoughtful question. Fewer training steps are harmful: with fixed batch size, performance vs. steps is non‑linear and exhibits an “aha” jump around ~50–150 steps (akin to the transitions reported in DeepSeekMath [1]). Cutting steps before this window often misses the jump and yields markedly worse unified grounding‑plus‑reasoning performance. For smaller batch sizes, effects are intertwined with other factors such as the learning rate: in our experience, training with smaller batch sizes can reach similar end performance but requires more steps to saturate.
>
> Reference:
>
> [1] Shao, Zhihong, et al. "Deepseekmath: Pushing the limits of mathematical reasoning in open language models." arXiv preprint arXiv:2402.03300 (2024).
>
> **Response to question 3 on a typo**
>
> We thank the reviewer for catching this. It is a typo: “Qwen2.5‑VL 2B” should be “Qwen2.5‑VL‑3B,” the same base model reported in Section 4.1 Setup. We will correct this in the next revision and ensure consistency throughout the paper and appendix.

---

> > ### Comment · Reviewer_zwCQ · 2025-08-03
> >
> > Thank you for your responses, which have addressed most of my concerns.
> >
> > Still, could you share quantitative results on how reducing training steps and batch size impacts model performance?

---

> > > ### Author Response · Authors · 2025-08-05
> > >
> > > Thanks for the comments. To address your request, we include below additional experiment results that illustrate how GRIT training results varies under reduced training steps and smaller batch sizes. All experiments use the same training configuration and hyperparameters as the original GRIT setup, and are conducted on the Qwen2.5-VL 3B model. The only variables changed are the total number of training steps and the total batch size.
> > >
> > > - Effect of training steps:
> > >  As shown in the first four rows, increasing the number of training steps from 50 to 200 results in consistent improvements overall in both answer accuracy (ACC) and grounding IoU (GIoU) across all evaluation tasks. Performance increases notably between 50–150 steps and stabilizes near 200 steps.
> > >
> > > - Effect of batch size:
> > >  Reducing the batch size to 8 while keeping the number of steps fixed (e.g., 50, 150, or 200) leads to noticeably weaker performance. This is expected, as smaller batch sizes under fixed steps yield fewer total training epochs. However, when training is extended to 3200 steps, matching the total number of epochs from the original setting, the performance recovers.
> > >
> > > | GRIT Training Setting                 | VSR ACC | VSR GIoU | TallyQA ACC | TallyQA GIoU | GQA ACC | GQA GIoU | MathVista ACC | MME ACC | OVD GIoU |
> > > |--------------------------------------|---------|----------|--------------|----------------|---------|-----------|----------------|----------|-----------|
> > > | Batch size = 128, steps = 50         | 63.8    | 0.308    | 41.2         | 0.305          | 55.0    | 0.420     | 48.8           | 82.5     | 0.271     |
> > > | Batch size = 128, steps = 100        | 59.6    | 0.341    | 45.5         | 0.433          | 54.1    | 0.501     | 49.6           | 79.4     | 0.390     |
> > > | Batch size = 128, steps = 150        | 72.8    | 0.380    | 47.7         | 0.427          | 62.1    | 0.478     | 56.1           | 85.3     | 0.387     |
> > > | Batch size = 128, steps = 200 (original) | 72.9    | 0.325    | 47.8         | 0.447          | 62.8    | 0.485     | 59.8           | 89.3     | 0.398     |
> > > | Batch size = 8, steps = 50           | 54.6    | 0.050    | 45.7         | 0.035          | 44.9    | 0.087     | 50.7           | 46.6     | 0.363     |
> > > | Batch size = 8, steps = 150          | 64.2    | 0.313    | 42.0         | 0.415          | 68.8    | 0.417     | 57.0           | 85.8     | 0.368     |
> > > | Batch size = 8, steps = 200          | 56.6    | 0.345    | 42.7         | 0.429          | 65.6    | 0.323     | 57.1           | 79.6     | 0.372     |
> > > | Batch size = 8, steps = 3200         | 72.2    | 0.328    | 45.7         | 0.440          | 69.2    | 0.462     | 57.5           | 92.9     | 0.406     |

---

> > > > ### Comment · Reviewer_zwCQ · 2025-08-05
> > > >
> > > > Thank you for the reply. I have no further questions.

---

### Official Review · Reviewer_7nZW · 2025-07-06

**Clarity:** 2
**Significance:** 2
**Originality:** 2
**Rating:** 5
**Confidence:** 3

**Summary:**

This paper introduces GRIT (Grounded Reasoning with Images and Texts), a novel framework designed to train multimodal large language models (MLLMs) to generate reasoning chains that are both coherent and visually grounded. Unlike prior work that relies solely on natural language to represent reasoning in vision-language tasks, GRIT enables models to interleave textual explanations with explicit bounding box references, thereby making the reasoning process more interpretable and grounded in visual input.

**Questions:**

1. **Clarification of Acronyms and Terminology**

   * In line 8, the acronym *MLLM* (Multimodal Large Language Model) is used without first introducing its full form. Similarly, *GRPO-GR* in line 13 is not expanded on first use. For clarity and accessibility, please ensure that all acronyms are defined when first introduced.
   * Additionally, it would be helpful to provide brief definitions for key technical terms such as *"grounding"*, particularly for readers from broader machine learning communities who may not be deeply familiar with vision-language grounding terminology.

2. **Details on Dataset Sampling and Test Splits**

   * The paper mentions that evaluation is conducted on data sampled from seven datasets. Could the authors clarify how these samples were selected (e.g., random, stratified) and how many examples from each dataset were used for testing? This would help readers better assess the diversity and robustness of the evaluation protocol.

3. **Missing Appendix or Supplementary Material**

   * The main text (e.g., lines 186, 210, and 215) references an Appendix, but no Appendix or supplementary material was found in the submission. Please clarify whether this content is missing or was inadvertently excluded.

**Ethical Concerns:**

["NO or VERY MINOR ethics concerns only"]

**Final Justification:**

The author's response resolved most of the issues and the scores were appropriately adjusted.

**Limitations:**

* The proposed training method relies on the evaluation of another MLLM (e.g., a GPT-based model) to compute the answer-accuracy reward. This dependency may introduce bias or circular evaluation, and raises questions about the method’s independence and generalizability.

* While GRIT demonstrates effectiveness on tasks such as object counting and spatial reasoning, it remains unclear how well the approach scales to more complex reasoning tasks (e.g., multi-step inference, causal reasoning, or abstract visual questions). Exploring these directions could be a valuable avenue for future work.

**Paper Formatting Concerns:**

Nothing

**Quality:**

3

**Strengths And Weaknesses:**

**Strengths:**

- The integration of textual explanations with explicit bounding box references enhances the interpretability and expandability of MLLMs, providing more transparent and grounded reasoning.
- The evaluation is conducted on test sets sampled from seven different datasets, showcasing the model’s generality across diverse vision-language tasks.
- The paper investigates the effect of scaling the training data and demonstrates that the proposed approach achieves strong performance with very limited supervision, highlighting its data efficiency.

**Weaknesses:**

- The proposed GRPO-GR method is more of an incremental modification—adding new reward terms to the existing GRPO algorithm—rather than a fundamentally novel algorithmic contribution.
- Even with 7,000 training samples, the model's in-domain accuracy remains below 64%, which may be insufficient for practical deployment in real-world applications.

---

> ### Author Rebuttal · Authors · 2025-07-30
>
> **Response to contribution of GRPO‑GR**
>
> We thank the reviewer for this perspective. In this work, we mainly propose the GRIT method and demonstrate that MLLMs trained with GRIT successfully unify grounding and reasoning abilities. Within GRIT, GRPO‑GR is one component of the contribution alongside the other one, grounded‑reasoning paradigm, leading to model outputs with interleave natural language with explicit bounding‑box coordinates. GRPO-GR is not a brand‑new optimization method, but rather an algorithm with new optimization target that is tailored to this grounded‑reasoning paradigm, i.e., rewards that enforce valid interleaved text+box structure and tie rationales to evidence, thereby enabling small MLLMs to “think with images” from answer‑level supervision alone.
>
> **Response to adequacy of accuracy for practical deployment**
>
> We thank the reviewer for raising this concern. Our paper’s focus is on unifying grounding and reasoning with training‑data efficiency for MLLMs. The in‑domain accuracy in Fig. 6 only reflects results under our intentionally constrained setting (3B backbone model, no domain‑specific warm‑start, 7k data), where the experiment aims at showing the importance of in-domain data and data quantity. Real‑world deployment entails greater robustness requirements and design choices such as model size and cold‑start initialization; we leave a full deployment study to future work.
>
>
> **Response to question 1 on undefined acronyms and terminology**
>
> We thank the reviewer for the clarity suggestion. We will define MLLM (Multimodal Large Language Model), GRPO‑GR (Group Relative Policy Optimization-Grounded Reasoning), and core terms such as grounding (the task of localizing specific objects or regions within an image based on a given textual description), at first mention and ensure consistent use throughout the paper and figures for accessibility.
>
> **Response to question 2&3 on experiment details and appendix**
>
> We appreciate the request for specifics and apologize for the missing information. For our evaluation, we curated testing data derived from six public open‑source datasets covering a range of visual reasoning and grounding tasks. The statistic for the testing data is shown in the following table. The appendix with these information was inadvertently omitted in the initial PDF export and we will further update our paper with these information.
>
> | Data source | VSR | TallyQA | GQA | MathVista | MME | OVDEval |
> |---|---:|---:|---:|---:|---:|---:|
> | **Counts** | 288 | 491 | 509 | 1000 | 240 | 2164 |
> | **Avg question/answer length** | 6.7 / 1.0 | 6.0 / 1.0 | 7.1 / 1.0 | 38.2 / 1.2 | 13.3 / 1.0 | 16.4 / 4 |
> | **Ratio of multi‑choice and yes/no questions (%)** | 71.2 | 0 | 58.9 | 70.8 | 100 | 0 |
> | **Ratio of annotated grounding targets (%)** | 58.8 | 25.6 | 25.3 | – | – | 17.3 |
>
>
>
> - VSR tests spatial relation verification. For our VSR evaluation set, we source question‑image‑answer triplets from the VSR subset of the Visual CoT benchmark [1] and manually filter out those with ambiguous answers, reducing the size from 404 to 288.
>
> - TallyQA focuses on counting; we uniformly sample evaluation questions where the target object counts range from 0 to 9 to create our TallyQA evaluation set.
>
> - GQA offers scene‑graph‑grounded, compositional object spatial questions. We first take the GQA subset from the Visual CoT benchmark [1] and then manually filter these to retain high‑quality instances for our GQA evaluation set.
>
> - From MME, we use only the counting, position, and existence subsets to broaden our evaluation scope.
>
> - MathVista evaluates mathematical reasoning in visual contexts. Following prior works, we adopt its *TestMini* split.
>
> - Finally, OVDEval is an open‑vocabulary detection (OVD) testing set that requires the model to ground fine‑grained semantics from the language query to the coordinates of visual features. We use its *position* subset and simplify it to object detection tasks with a single target existed in the target image.
>
> Reference:
> [1] Shao, Hao, et al. "Visual cot: Advancing multi-modal language models with a comprehensive dataset and benchmark for chain-of-thought reasoning." Advances in Neural Information Processing Systems 37 (2024): 8612-8642.
>
>
>
> **Response to the limitation suggestions**
>
> We appreciate the reviewer’s insightful suggestions. We would like to point out that while our approach uses a GPT-based model as a reward judge, it is only employed to compare a candidate answer against a ground truth answer, which limits (though does not fully eliminate) potential bias; we will further discuss and clarify this in the revised manuscript. We also acknowledge that our current focus on teaching MLLMs to think with images leaves open questions about scaling to more complex reasoning tasks, which we will explicitly highlight as an important direction for future work.

---

> > ### Comment · Reviewer_7nZW · 2025-08-06
> >
> > Thanks for your response. I have no other questions. I will raise the rating accordingly.

---

### Official Review · Reviewer_5Svj · 2025-07-06

**Clarity:** 4
**Significance:** 3
**Originality:** 3
**Rating:** 5
**Confidence:** 5

**Summary:**

This paper proposes GRIT, a framework using GROP to train VLMs to enable models with the grounded reasoning capabilities, where the model outputs grounding boxes during reasoning. Extensive experiments show that the proposed methods can improve the performance of VLMs on various benchmarks effectively.

**Questions:**

Will the Visual Grounding task (e.g., asking more complex compositional questions) benefit from this grounded reasoning method?

**Ethical Concerns:**

["NO or VERY MINOR ethics concerns only"]

**Final Justification:**

Most of my concerns have been addressed. I will keep my positive score.

**Limitations:**

yes

**Quality:**

3

**Strengths And Weaknesses:**

Strengths
- An empirically reliable method to improve the performance of VLMs, and in the meantime to incorporate reasoning evidence during output.
- Extensive experiments have conducted to investigate the results.

Weakness
- The proposed method leverages grounded evidence to improve the reasoning accuracy of VLMs. However, no experiments are conducted to show if this method can also facilitate the improvements on Visual Grounding task.

Missing References:
- Wang W, Lv Q, Yu W, et al. Cogvlm: Visual expert for pretrained language models[J]. Advances in Neural Information Processing Systems, 2024, 37: 121475-121499.
- Qi J, Ding M, Wang W, et al. Cogcom: A visual language model with chain-of-manipulations reasoning[C]//ICLR. 2025.

---

> ### Author Rebuttal · Authors · 2025-07-30
>
> **Response to generalizability of GRIT to visual grounding tasks**
>
> We thank the author for raising this point. Our paper’s primary objective is to show that GRPO‑GR can teach small MLLMs to “think with images” using tiny data, requiring both reasoning and visual grounding abilities. In the Experiments section, we also evaluated models in a visual-grounding‑oriented setting: OVDEval, which requires detecting locations of specified targets. As noted in the paper (line 252), GRIT‑trained models achieve even more accurate results on OVDEval than zero‑shot MLLMs. Crucially, these gains are achieved with only VQA‑style answer supervision—no ground‑truth grounding labels—indicating emergent improvements in visual grounding. That said, we believe that with more specific training data in the visual grounding domain, GRIT is well‑suited to further benefit models in visual grounding tasks as the proposed optimization target can be directly applied. We are making plans to further explore along this direction.
>
> **Response to missing related work**
>
> We thank the reviewer for pointing out these references. We will add and discuss CogVLM (NeurIPS’24) and CogCoM (ICLR’25), and clarify how GRIT differs: our method learns interleaved text‑and‑bounding-box reasoning traces via RL, without requiring chain or box supervision, directly targeting the capability to “think with images.”
>
>
> **Response to the question of benefit on compositional visual grounding**
>
> As GRIT is designed to teach MLLMs with interleaving language with explicit, self‑selected boxes, we believe it encourages tighter text‑region coupling, which is conducive to compositional grounding. Our experiments on the models’ grounding performance (the GIoU metrics in Table 1) also suggest this transfer, where GRIT‑trained models achieve stronger grounding performance than zero‑shot MLLMs even though training uses only visual-question answering data. In the future revision, we are planning to include demonstrations on how GRIT can be applied to training models for compositional grounding tasks with more in-domain training data.

---

> > ### Comment · Reviewer_5Svj · 2025-08-08
> >
> > Thanks for the response provided by the authors. I will keep my score as positive.

---

### Note · Authors · 2025-08-13

Dear Chairs and Reviewers,

We are sincerely grateful for your thoughtful reviews, constructive suggestions, and engagement during the rebuttal and discussion phases. We are encouraged that all three reviewers maintained positive recommendations, highlighting GRIT’s contribution in unifying grounding and reasoning abilities in MLLMs, its data efficiency—achieving strong performance with as few as 20 training examples—and its transparent, interpretable outputs through interleaved text–bounding-box reasoning chains.

Reviewers recognized our GRIT method as an empirically reliable approach (Reviewer 5Svj), noted its efficiency and effectiveness to bridge textual reasoning with visual evidence (Reviewer zwCQ and 7nZW), and valued the broad evaluation across diverse tasks and datasets (all three reviewers).

During the rebuttal, we addressed all raised concerns in detail, and some highlights are:

- Generalizability to various grounding tasks. We clarified that GRIT already improves grounding performance in OVDEval without explicit box supervision and is well-suited for various visual grounding tasks, including compositional grounding, with more domain-specific data added.
- Clarifications on terminology and evaluation details. We explained terms for broader audiences, and provided the full statistics and sampling protocols for our evaluation datasets.
- Scalability to larger models. We explained that GRIT is model-agnostic; gains on small backbones expected to transfer; large-model eval planned when resources allow.
- Further exploration on training hyperparameters. We examined the impact of training steps and batch size, finding a clear “aha” jump in performance between ~50–150 steps; fewer steps hurt results, and small batches under fixed steps perform worse but recover when trained longer to match total epochs.

We are grateful that all reviewers acknowledged our clarifications and one reviewer explicitly mentioning further raising the score. We believe the final version of our work demonstrates a robust, general, and efficient method for teaching MLLMs to “think with images,” with broad potential impact on vision-language reasoning research.

Thank you for your time, constructive input, and careful consideration.

Best regards,

The Authors

---

### Decision · Program_Chairs · 2025-09-17

**Decision:**

Accept (poster)

**Comment:**

The paper introduces a reinforcement learning approach for training a VLM to generate reasoning traces that interleave natural language with bounding boxes. While some concerns were initially raised regarding the empirical studies, the authors addressed these effectively in the rebuttal. As the reviewers are generally consistent in their positive assessment, I recommend acceptance.